# Transport of orbital currents in systems with strong intervalley coupling: the case of Kekulé distorted graphene

Tarik P. Cysne,[1, *] R. B. Muniz,[1] and Tatiana G. Rappoport[2, 3]

[1]*Instituto de Física, Universidade Federal Fluminense, 24210-346 Niterói RJ, Brazil*
[2]*Physics Center of Minho and Porto Universities (CF-UM-UP), Braga, Portugal*
[3]*Instituto de Física, Universidade Federal do Rio de Janeiro,*
*C.P. 68528, 21941-972 Rio de Janeiro RJ, Brazil*
(Dated:)

We show that orbital currents can describe the transport of orbital magnetic moments of Bloch states in models where the formalism based on valley current is not applicable. As a case study, we consider Kekulé-$O$ distorted graphene. We begin by analyzing the band structure in detail and obtain the orbital magnetic moment operator for this model within the framework of the modern theory of magnetism. Despite the simultaneous presence of time-reversal and spatial-inversion symmetries, such operator may be defined, although its expectation value at a given energy is zero. Nevertheless, its presence can be exposed by the application of an external magnetic field. We then proceed to study the transport of these quantities. In the Kekulé-$O$ distorted graphene model, the strong coupling between different valleys prevents the definition of a bulk valley current. However, the formalism of the orbital Hall effect together with the non-Abelian description of the magnetic moment operator can be directly applied to describe its transport in these types of models. We show that the Kekulé-$O$ distorted graphene model exhibits an orbital Hall insulating plateau whose height is inversely proportional to the energy band gap produced by intervalley coupling. Our results strengthen the perspective of using the orbital Hall effect formalism as a preferable alternative to the valley Hall effect approach.

## I. INTRODUCTION

The manipulation of electronic orbital angular momentum (OAM) has garnered increasing interest in recent years, giving rise to the emerging field of orbitronics [1–9]. One of the cornerstones of this field is the so-called orbital Hall effect (OHE) that allows the generation of orbital currents in systems with negligible spin-orbit coupling [10–23]. This effect has also gained importance beyond pragmatic applications in orbitronics, allowing for a deeper understanding of some topological aspects of multi-orbital systems [24–29]. Additionally, the OHE may shed light on some long-standing discussions surrounding the valley Hall effect (VHE) and the definition of valley currents.

The VHE is built upon the concept of valley current [30], which in the case of regular graphene involves two inequivalent valleys $\mathbf{K}$ and $\mathbf{K}'$. In this case, the valley current density may be defined by

$$\mathbf{J}_v = (-e)\left(\mathbf{v}|_{\mathbf{K}} - \mathbf{v}|_{\mathbf{K}'}\right), \qquad (1)$$

where, $\mathbf{v}|_{\mathbf{K}}$ and $\mathbf{v}|_{\mathbf{K}'}$ are the valley-projected velocity operators. The concepts underlying the VHE have indeed contributed to the understanding of important electronic transport properties in condensed matter [31–35]. Nevertheless, in recent theoretical and experimental studies, some uncertainties surrounding these concepts have come to light. [36–42], as reviewed in Ref. [43]. The valley quantum number itself does not couple with experimental probing fields. It is the magnetic moment associated with valleys that may be observed. Part of the puzzle associated with the physics of the VHE may be related to the confusion between the magnetic moment operator and its expectation value. The Bloch states orbital magnetic moment operator was clearly delineated in the original works within the modern theory of magnetism. For degenerate bands, it acquires a matrix structure [31, 32, 44]. However, misleading notions from earlier studies suggest that the simultaneous presence of spatial inversion and time-reversal symmetries impedes the transport of magnetic moments. In reality, these symmetries merely confine the existence of finite magnetic moment expectation value in equilibrium. As we elucidate in this paper, by employing a straightforward model that adheres to these symmetries, the transport of magnetic moments is not forbidden, despite their expectation value vanishing at any point in $k$-space in equilibrium.

Recently, Bhowal and Vignale [45] proposed that the OHE may offer an effective and clear alternative description to the physics underlying the VHE. In this view, the magnetic moment associated with the valleys would be transported by an orbital current density defined by

$$\mathbf{J}^{L_z} = \frac{1}{2}\left\{\mathbf{v}, \hat{L}_z\right\}, \qquad (2)$$

where $\hat{L}_z$ is the OAM operator associated with Bloch states magnetic moments. Orbital current yields different outcomes in orbital transport compared to valley current. For instance, the existence of the OHE in centrosymmetric and non-magnetic systems follows naturally [25, 46]. One advantage of this description is to avoid the need for a valley projection typically used to define valley current.

* tarik.cysne@gmail.com

arXiv:2404.12072v1 [cond-mat.mes-hall] 18 Apr 2024

The need for valley projection within the valley Hall perspective imposes serious limitations when dealing with models with strong valley mixing.

In this work, we illustrate the flexibility of using orbital current by studying the OHE in a model that respects spatial inversion and time-reversal symmetries and shows strong intervalley mixing, v.i.z., the Kekulé-distorted graphene of type O [47–50]. As will be pedagogically presented below, the calculation of the orbital Hall conductivity in this type of model follows straightforwardly from the approach introduced in Refs. [45, 46].

## II. KEKULÉ DISTORTED GRAPHENE

The electronic structure of the regular graphene monolayer is characterized by two gapless Dirac cones located at inequivalent valleys of the Brillouin zone (BZ) $\mathbf{K}$ and $\mathbf{K}' = -\mathbf{K}$. At low-energy regime, the electronic properties are described by the Dirac Hamiltonian,

$$\mathcal{H}_g(\tilde{\mathbf{q}}) = \hbar v_\mathrm{F}(\tau_z \sigma_x \tilde{q}_x + \sigma_y \tilde{q}_y), \tag{3}$$

where, $\sigma_{x,y,z}$ are Pauli matrices related to the sublattice degree of freedom of graphene. $\tau_z = \mathrm{diag}(1,-1)$ is the Pauli matrix related to the valley degree of freedom. $\tilde{q}_{x,y}$ are cristal momentum measured with respect to the valleys and $v_\mathrm{F}$ is the Fermi velocity.

In this work, we consider a graphene monolayer subjected to a Kekulé-type distortion. These distortions can be achieved by inducing a spatial perturbation with a periodicity of $\mathbf{G} = (\mathbf{K}_+ - \mathbf{K}_-)$ [see Fig. 1 (a)]. This leads to a BZ folding and shifts of the low-energy sector to the $\Gamma$-point of the supercell BZ [Fig 1 (b)]. There are two types of Kekulé distortion: Y-distortion (Kek-Y) and O-distortion (Kek-O). The Kek-Y distortion preserves graphene´s gapless structure and modifies its Fermi velocity. This phenomenon was experimentally observed in graphene grown on top of copper substrate [51]. The Kek-O distortion is particularly interesting for our purposes. In this scenario, the BZ folding is accompanied by a bandgap opening. This distortion has been recently observed in graphene intercalated with lithium [50] and also has been predicted to occur in graphene grown on top of some topological insulators [52]. We follow Refs. [47–49] and choose the basis $\beta_\mathrm{Kek} = \{\Psi_{\mathbf{K}'}^\mathrm{T}, \Psi_{\mathbf{K}}^\mathrm{T}\} = \{-|B,\mathbf{K}'\rangle, |A,\mathbf{K}'\rangle, |A,\mathbf{K}\rangle, |B,\mathbf{K}\rangle\}$, where the superscript T means the transpose operation and $A, B$ represents the two distinct graphene sublattices. Thus, on this basis, the low-energy Hamiltonian of the Kek-O distorted graphene can be cast as

$$\mathcal{H}_O(\mathbf{q}) = \begin{bmatrix} 0 & \gamma_- & \Delta & 0 \\ \gamma_+ & 0 & 0 & -\Delta \\ \Delta & 0 & 0 & \gamma_- \\ 0 & -\Delta & \gamma_+ & 0 \end{bmatrix}, \tag{4}$$

where, $\gamma_\pm = \hbar v_\mathrm{F}(q_x \pm i q_y) = \hbar v_\mathrm{F} q e^{\pm i\phi}$. Here $q_{x,y}$ is the crystal momentum measured from the Kek-O BZ center

Fig. 1 (b). The term $\Delta$ arises from Kek-O distortion, resulting in a coupling between sectors $\Psi_{\mathbf{K}'}$ and $\Psi_{\mathbf{K}}$ of the Hilbert space. The functional dependency of $\Delta$ with the parameters of the original lattice Hamiltonian can be found in Ref. [47–49]. It is worth noting that in the distortion represented in Fig. 1 (a), no point group symmetry is broken. The superlattice structure only reduces translational symmetry. In particular, Kek-O distorted graphene preserves spatial inversion symmetry [53, 54].

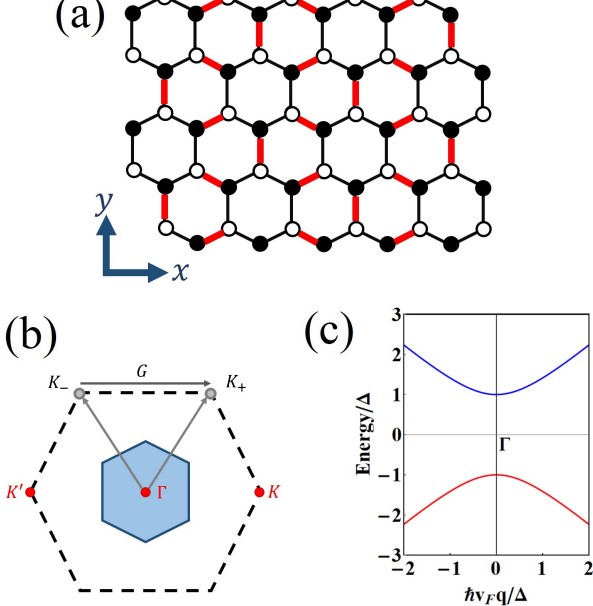

Figure 1. (a) Kekulé distorted honeycomb lattice of type O (Kek-O). The black lines indicate slightly weaker (longer) bonds, while the red lines indicate slightly stronger (shorter) bonds. (b) The Brillouin zones of regular graphene (dashed black) and Kekulé-distorted graphene (blue). (c) Electronic spectra of Kek-O distorted graphene around the $\Gamma$ point.

### A. Eigenvectors and Energy Spectra

It is straightforward to compute the energy spectrum of the Hamiltonian given by Eq. (4). One obtains a doubly degenerate valence energy band,

$$E_{v,n}(\mathbf{q}) = -\sqrt{\hbar^2 v_\mathrm{F}^2 q^2 + \Delta^2} = -\varepsilon(\mathbf{q}), \tag{5}$$

and a doubly degenerate conduction energy band,

$$E_{c,n}(\mathbf{q}) = +\sqrt{\hbar^2 v_\mathrm{F}^2 q^2 + \Delta^2} = +\varepsilon(\mathbf{q}), \tag{6}$$

where $n = 1, 2$ for each branch in the energy degenerate subspace.

One can calculate the eigenvectors of the energy degenerate subspaces and enforce their orthonormality. Thus,

for the valence band, the eigenvectors are given by,

$$|u_{v,1}(\mathbf{q})\rangle = \left[ -\frac{\gamma_-}{\sqrt{2}\varepsilon(\mathbf{q})}, \frac{1}{\sqrt{2}}, 0, \frac{\Delta}{\sqrt{2}\varepsilon(\mathbf{q})} \right]^{\mathrm{T}}, \tag{7}$$

$$|u_{v,2}(\mathbf{q})\rangle = \left[ -\frac{\Delta}{\sqrt{2}\varepsilon(\mathbf{q})}, 0, \frac{1}{\sqrt{2}}, -\frac{\gamma_+}{\sqrt{2}\varepsilon(\mathbf{q})} \right]^{\mathrm{T}}, \tag{8}$$

and for the conduction band,

$$|u_{c,1}(\mathbf{q})\rangle = \left[ -\frac{\gamma_-}{\sqrt{2}\varepsilon(\mathbf{q})}, -\frac{1}{\sqrt{2}}, 0, \frac{\Delta}{\sqrt{2}\varepsilon(\mathbf{q})} \right]^{\mathrm{T}}, \tag{9}$$

$$|u_{c,2}(\mathbf{q})\rangle = \left[ \frac{\Delta}{\sqrt{2}\varepsilon(\mathbf{q})}, 0, \frac{1}{\sqrt{2}}, \frac{\gamma_+}{\sqrt{2}\varepsilon(\mathbf{q})} \right]^{\mathrm{T}}. \tag{10}$$

It is important to note that when $\mathbf{q} \to 0$ ($\gamma_\pm \to 0$, $\varepsilon(\mathbf{q}) \to |\Delta| \neq 0$ ), all eigenstates exhibit a superposition of equal-weighted amplitudes associated to valleys $\mathbf{K}$ and $\mathbf{K}'$. The states also present maximum entanglement between valley and sublattice. In contrast to graphene with sublattice potential, it is impossible to decouple the Hilbert space for the Kekulé distorted graphene into two subspaces with well-defined valley quantum numbers. Hence, it is meaningless to use Eq. (1) to define a valley current in the case of Kekule-distorted graphene Hamiltonian [Eq. (4)].

## III. BLOCH STATES ORBITAL MAGNETIC MOMENT

The existence of Bloch state orbital magnetic moment was first derived by Kohn using perturbation theory [55] and later re-obtained through the application of semi-classical wave packet formalism [32, 56, 57]. In this formalism, it may be interpreted as the self-rotation of the wave packet. For the case of degenerate bands, the Bloch state orbital magnetic moment acquires a non-Abelian (matrix) structure [31, 32, 44]. This non-Abelian structure permits the appearance of non-zero matrix elements of orbital magnetic moments, even in systems that respect both spatial inversion and time-reversal symmetry, e.g., bilayer transition-metal dichalcogenides [see Appendix of Ref. [46]].

In the case of Kek-O graphene, we have two degenerate subspaces, each with dimension 2. One formed by the conduction band, and the other by the valence band. Within each subspace, the matrix elements of the orbital magnetic moment are

$$m_{b;n,m}^{z,u}(\mathbf{q}) = \left( \frac{e}{2\hbar} \right) \cdot$$
$$\mathrm{Im} \left[ \langle \vec{\nabla}_{\mathbf{q}} u_{b,n}(\mathbf{q}) | \times \left( \mathcal{H}_O(\mathbf{q}) - \tilde{E}_{b,m,n}(\mathbf{q})\hat{\mathbb{1}} \right) | \vec{\nabla}_{\mathbf{q}} u_{b,m}(\mathbf{q}) \rangle \right], \tag{11}$$

where, $b = c, v$ and the sub-indexes $n$ and $m = 1, 2$. $\vec{\nabla}_{\mathbf{q}} = \hat{x}\partial/\partial q_x + \hat{y}\partial/\partial q_y$ and $\times$ represents the cross-product. Also, we define $\tilde{E}_{b,m,n}(\mathbf{q}) = (E_{b,n}(\mathbf{q}) + E_{b,m}(\mathbf{q})) /2$. In the valence band subspace ($b = v$), one uses the energy given by Eq. (5) and $n, m = 1, 2$ designate eigenvectors of Eqs. (7) and (8). Similarly, in the conduction band subspace ($b = c$), one uses the energy given by Eq. (6), with $n, m = 1, 2$ designating eigenvectors of Eqs. (9) and (10). The matrix elements coupling conduction and valence bands are disregarded in the non-Abelian formulation as detailed in Refs. [31, 32, 44]. After straightforward calculations one obtains

$$\hat{\mathbb{m}}^{z,u}(\mathbf{q}) = \begin{bmatrix} \hat{\mathbb{m}}_v^{z,u}(\mathbf{q}) & \mathbb{0}_{2\times 2} \\ \mathbb{0}_{2\times 2} & \hat{\mathbb{m}}_c^{z,u}(\mathbf{q}) \end{bmatrix} = \frac{2m_0(\mathbf{q})}{\sqrt{\Delta^2 + q^2\hbar^2 v_{\mathrm{F}}^2}} \begin{bmatrix} -\Delta & \hbar v_{\mathrm{F}} q e^{i\phi} & 0 & 0 \\ \hbar v_{\mathrm{F}} q e^{-i\phi} & \Delta & 0 & 0 \\ 0 & 0 & \Delta & \hbar v_{\mathrm{F}} q e^{i\phi} \\ 0 & 0 & \hbar v_{\mathrm{F}} q e^{-i\phi} & -\Delta \end{bmatrix}, \tag{12}$$

where $m_0(\mathbf{q}) = \left( \frac{e}{\hbar} \right) \frac{\Delta \hbar^2 v_{\mathrm{F}}^2}{2\left( \Delta^2 + q^2\hbar^2 v_{\mathrm{F}}^2 \right)}$. This matrix is written on the eigenstates basis $\beta_u = \{|u_{v,1}\rangle, |u_{v,2}\rangle, |u_{c,1}\rangle, |u_{c,2}\rangle\}$.

At this point, it is instructive to consider the impact of an external magnetic field on the electronic spectra of distorted graphene. For weak intensities of the external magnetic field ($B < 1$ T), the energy is corrected by a factor proportional to the diagonal elements of the matrix

of Eq. (12):

$$E_{b,n}^B(\mathbf{q}) = E_{b,n}(\mathbf{q}) - B \cdot m_{b;n,n}^{z,u}(\mathbf{q})$$
$$= E_{b,n}(\mathbf{q}) \pm \frac{2\Delta m_0(\mathbf{q}) \cdot B}{\sqrt{\Delta^2 + \hbar^2 v_{\mathrm{F}}^2 q^2}}, \tag{13}$$

where the sign $+$ is for states $|u_{v,1}\rangle, |u_{c,2}\rangle$ and the sign $-$ for states $|u_{v,2}\rangle, |u_{c,1}\rangle$. As illustrated in Fig. 2, the external magnetic field lifts the energy degeneracy near $\Gamma$ exposing the orbital magnetic moment associated with

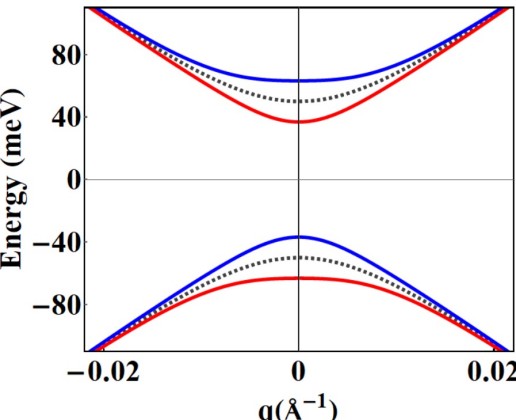

Figure 2. Effect of the external magnetic field in the low-energy spectra of Kek-O distorted graphene. The dashed gray curve shows the low-energy spectra near the $\Gamma$ point of the folded BZ without an applied magnetic field for $\Delta = 50$meV. The blue $(E_{v,1}^B, E_{c,2}^B)$ and red $(E_{v,2}^B, E_{c,1}^B)$ curves show the electronic spectrum for an applied external magnetic field $B = 1.0$ T.

Bloch bands of the Hamiltonian. This can originate a giant Zeeman effect for small values of the Kekulé distortion.

Figure 2 depicts the Zeeman shift induced by the external magnetic field. The dotted lines represent the energy spectrum for $B = 0$, while the red and blue curves illustrate the energy spectrum for $B \neq 0$ and opposite orbital magnetic moments. In contrast to graphene with sublattice asymmetry or TMD layers [58], the relative shift of states with opposite orbital magnetic moments cannot be attributed to individual valleys, as each state is a linear combination of valley states.

To obtain the orbital magnetic moment operator in the $\beta_{\text{Kek}}$ basis, which is useful for the transport calculations, one needs first to define a unitary transformation $U(\mathbf{q})$ that changes the basis from $\beta_u$ to $\beta_{\text{Kek}}$. Such an operator obeys the relation $\hat{U}^\dagger(\mathbf{q})\mathcal{H}_O(\mathbf{q})\hat{U}(\mathbf{q}) = \text{diag}\left[E_{v,1}(\mathbf{q}), E_{v,2}(\mathbf{q}), E_{c,1}(\mathbf{q}), E_{c,2}(\mathbf{q})\right]$ and can be easily constructed from eigenvectors of Eqs. (7-10). Thus, applying it to the matrix of Eq. (12), one obtains

$$\hat{\mathbb{m}}^{z,\text{Kek}}(\mathbf{q}) = \hat{U}(\mathbf{q})\hat{\mathbb{m}}^{z,u}(\mathbf{q})\hat{U}^\dagger(\mathbf{q})$$
$$= \begin{bmatrix} 0 & 0 & -m_0(\mathbf{q}) & 0 \\ 0 & 0 & 0 & -m_0(\mathbf{q}) \\ -m_0(\mathbf{q}) & 0 & 0 & 0 \\ 0 & -m_0(\mathbf{q}) & 0 & 0 \end{bmatrix}. (14)$$

The existence of such an operator allows the orbital transport when the system is out of equilibrium. This occurs despite the general belief that the simultaneous occurrence of spatial-inversion symmetry and time-reversal symmetry imposes the absence of magnetic moment transport. This arises from neglecting the non-Abelian nature of the magnetic moment in nearly-degenerate bands [46].

## IV. ORBITAL CURRENT OPERATOR

To study orbital transport, we follow Refs. [45, 46] and define an OAM operator using the matrix from the Eq.(14):

$$\hat{L}_z(\mathbf{q}) = -(\hbar/\mu_B g_L)\hat{\mathbb{m}}^{z,\text{Kek}}(\mathbf{q}). \qquad (15)$$

where, $\mu_B = e\hbar/(2m_e)$ is the Bohr magneton in terms of the electron rest mass $m_e$ and $g_L = 1$ is the Landé g-factor. In general, for multi-orbital systems like transition metal dichalcogenides [46], this operator takes into account contributions from both intrasite (intra-atomic) electron motion and intersite movement [59, 60]. In the case of Kek-O graphene, the band structure is governed by orbitals $p_z$, which lack intra-atomic contributions. Thus, Eq. (15) is ruled by the intersite movement of electrons, similarly to the modern theory of orbital magnetization [61, 62].

One can use Eq. (2) to define an orbital current density associated with the operator defined in Eq. (15) without any ambiguity. The velocity operators $\hat{v}_{x(y)}(\mathbf{q}) = \hbar^{-1}\partial\mathcal{H}_O(\mathbf{q})/\partial q_{x(y)}$ can be easily calculated from the Hamiltonian of Eq. (4). With the use of Eqs. (15) and (2) we obtain

$$\hat{J}_x^{L_z} = \frac{v_F m_0(\mathbf{q})\hbar}{\mu_B} \begin{bmatrix} 0 & 0 & 0 & -i \\ 0 & 0 & i & 0 \\ 0 & -i & 0 & 0 \\ i & 0 & 0 & 0 \end{bmatrix}, \qquad (16)$$

$$\hat{J}_y^{L_z} = \frac{v_F m_0(\mathbf{q})\hbar}{\mu_B} \begin{bmatrix} 0 & 0 & 0 & 1 \\ 0 & 0 & 1 & 0 \\ 0 & 1 & 0 & 0 \\ 1 & 0 & 0 & 0 \end{bmatrix}, \qquad (17)$$

which shall be used to calculate the OHE for the Kek-O distorted graphene.

## V. ORBITAL HALL EFFECT

In the previous section, we have shown that the orbital current density given by Eq. (2) is well-defined even for models that present strong valley mixing terms. Here, we use Eqs. (16, 17) to calculate the orbital Hall conductivity of the Kek-O distorted graphene and show that it presents an orbital Hall insulating phase. Considering that the system is subject to a driven electric field applied in the x-direction, an orbital current will flow in the y-direction. This phenomenon is governed by $\mathcal{J}_y^{L_z} = \sigma_{\text{OH}}^{L_z}\mathcal{E}_x$ where orbital Hall conductivity is given by,

$$\sigma_{\text{OH}}^{L_z} = e \sum_{n,b} \int \frac{d^2q}{(2\pi)^2} f_{n,b}(\mathbf{q})\Omega_{n,b}^{L_z}(\mathbf{q}), \qquad (18)$$

where $f_{n,b}(\mathbf{q}) = \Theta(E_F - E_{n,b}(\mathbf{q}))$ is the Fermi-Dirac distribution at zero temperature, $E_F$ is the Fermi energy, and $\Omega_{n,b}^{L_z}(\mathbf{q})$ represents the orbital-weighted Berry curvature given by

$$\Omega_{n,b}^{L_z}(\mathbf{q}) = 2\hbar \sum_{n' \neq n} \sum_{b' \neq b} \text{Im} \left[ \frac{\langle u_{n,b}(\mathbf{q})| \, \hat{v}_x(\mathbf{q}) \, |u_{n',b'}(\mathbf{q})\rangle \, \langle u_{n',b'}(\mathbf{q})| \, \hat{J}_y^{L_z}(\mathbf{q}) \, |u_{n,b}(\mathbf{q})\rangle}{\left(E_{n,b}(\mathbf{q}) - E_{n',b'}(\mathbf{q})\right)^2} \right]. \quad (19)$$

Using Eqs. (5-10) it is possible to obtain

$$\Omega_{n,v(c)}^{L_z}(\mathbf{q}) = \mp \left( \frac{e}{\hbar\mu_B} \right) \frac{\Delta^2 \hbar^4 v_F^4}{4 \left( \Delta^2 + q^2 \hbar^2 v_F^2 \right)^{5/2}}, \quad (20)$$

where, we use the $-$ and the $+$ signs for the valence and conduction bands, respectively. Substituting this expression in Eq. (18), we can find analytical expressions for orbital Hall conductivity. Writing $\sigma_{\text{OH}}^{L_z}(E_F) = \sum_{n,b} \sigma_{\text{OH}}^{n,b}(E_F)$, we obtain:

$$\sigma_{\text{OH}}^{n,v}(E_F) = g_s \left( \frac{e^2}{2\pi\hbar\mu_B} \right) \frac{\Delta^2 \hbar^2 v_F^2}{6 \left( \Delta^2 + q_{F,v}^2 \hbar^2 v_F^2 \right)^{3/2}} \quad (21)$$

and,

$$\sigma_{\text{OH}}^{n,c}(E_F) = -g_s \left( \frac{e^2}{2\pi\hbar\mu_B} \right)$$
$$\cdot \left( \frac{\hbar^2 v_F^2}{6\Delta} - \frac{\Delta^2 \hbar^2 v_F^2}{6 \left( \Delta^2 + q_{F,c}^2 \hbar^2 v_F^2 \right)^{3/2}} \right), (22)$$

where $g_s = 2$ is the spin-degeneracy. Here $q_{F,v(c)}$ denotes the Fermi momentum for the valence $(v)$ and conduction $(c)$ bands

$$q_{F,v} = \begin{cases} \sqrt{E_F^2 - \Delta^2}/(\hbar v_F) & \text{for } E_F < -\Delta \\ 0 & \text{otherwise} \end{cases} \quad (23)$$

and

$$q_{F,c} = \begin{cases} \sqrt{E_F^2 - \Delta^2}/(\hbar v_F) & \text{for } E_F > +\Delta \\ 0 & \text{otherwise} \end{cases} \quad (24)$$

Fig. 3 shows the orbital Hall conductivity calculated as a function of $E_F$ for different values of the Kek-O coupling $\Delta$. Particularly, in the dashed blue curve, we used the value of $\Delta$ obtained in the experiment reported in Ref. [50] using graphene intercalated with lithium atoms.

We note that when the $E_F$ lies inside the insulating bandgap, the orbital Hall conductivity shows a plateau with height

$$\bar{\sigma}_{\text{OH}}^{L_z} = \frac{\hbar^2 v_F^2}{6\Delta} \left( \frac{g_s e^2}{2\pi\hbar\mu_B} \right) = \frac{2}{3} \frac{\mu_B^*}{\mu_B} \left( \frac{e}{2\pi} \right), \quad (25)$$

where $\mu_B^* = (e\hbar)/(2m^*)$ in the second equality is an effective Bohr magneton, written in terms of the effective mass of Bloch electron $m^* = \Delta/v_F^2$. Eq. (25) coincides with results obtained in graphene endowed with sublattice potential [45] and transition metal dichalcogenides [46]. The height of the orbital Hall insulating plateau should be robust against dilute disorder due to the absence of a Fermi surface [18, 19, 63].

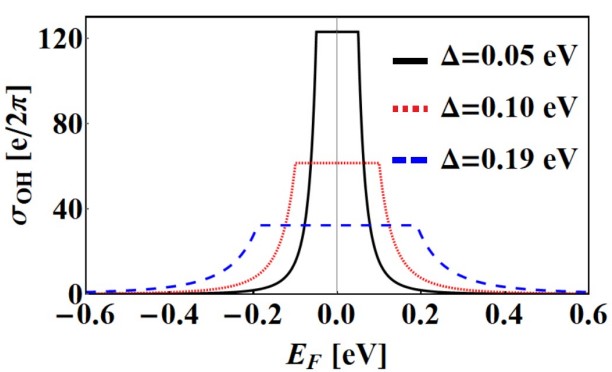

Figure 3. Orbital Hall conductivity calculated as a function of Fermi energy for distinct values of parameters $\Delta$. We used $\Delta = 0.05$ eV (black solid), $\Delta = 0.10$ eV (red dotted), and the value obtained in the experiment reported in Ref. [50], $\Delta = 0.19$ eV (blue dashed). The Fermi velocity of graphene is $v_F = 3at/(2\hbar)$, where $a = 1.42\text{Å}$ and $t = 2.8$ eV.

## VI. FINAL REMARKS AND CONCLUSIONS

We have demonstrated the unambiguous definition of an orbital current density operator for models featuring inter-valley coupling, using Kek-O distorted graphene as a case study. We also analytically obtained the orbital Hall insulator plateau for this model and examined its dependence on the intervalley mixing coupling term. Attempting a similar procedure using a valley current description would encounter challenges stemming from the intervalley coupling term, which produces eigenstates with significant superposition across different valleys. Although it is still possible to use a pseudo-spin defined in terms of a linear combination of valleys, the superposition would forbid simple valley-projection procedures commonly used in the valleytronic community and the definition of the sign of the associated magnetic moment is not straighforward.

Ref. [64] argues for the existence of the valley Hall insulator phase in the presence of an intervalley coupling term. However, it does not present the formal construction of the valley current operator or the calculation of the valley Hall conductivity. The valley-polarized transport in a system with intervalley coupling is carefully studied in Ref. [65]. However, the authors do not adopt a bulk perspective but instead use a device-based approach employing the transmission coefficient formalism. In this scenario, the difficulty associated with defining the valley current equation is overcome, albeit at the cost of losing the bulk description.

The possibility of studying orbital magnetic moment transport in models with strong intervalley mixing

strengthens the proposal that the orbital Hall approach is more suitable to describe this response than the VHE [45]. This may open the door to a review of the co-nundrum surrounding recent studies of the valleytronic community [43].

## ACKNOWLEDGMENTS

We acknowledge CNPq/Brazil, CAPES/Brazil, FAPERJ/Brazil. TGR acknowledges funding from FCT-Portugal through Grant No. 2022.07471.CEECIND/CP1718/CT0001 (https://doi.org/10.54499/2022.07471.CEECIND/CP1718/CT0001)

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
