# Peer review of "Transport of orbital currents in systems with strong intervalley coupling: the case of Kekulé distorted graphene"

_SciPost Physics Core_

## Round 1 · Referee Report · Anonymous (Referee 1) · 2024-5-13

Strengths

1- Analytical model, all calculations can be reproduced 2-Addresses a problem which is actively debated

Weaknesses

1-Interpretation - valley currents are meaningless in the considered situation
2-Experimental implications: how is the envisaged situation achieved in the lab - what are the main predictions of this work?

Report

The authors report a theoretical investigation of orbital currents in a Kekulé deformed graphene lattice, occurring when the couplings between carbon atoms are either enhanced or decreased (Figure 1a). How this particular situation is achieved in practice is not discussed in sufficient detail (doesn't the envisaged situation require a very careful fine-tuning?), and the model calculation has the risk of being "just a model" - not relevant to reality. (Later, the paper uses a value for the Kekulé parameter extracted from experiment, but the experiment is itself somewhat unclear.)
But my main point is philosophical. The band structure shown in Fig. 1 c does not have two valleys because the energy minimum is shifted to the Gamma-point - hence it does not make any sense to speak about "valley currents", which are specific to the K and K' points in the undistorted lattice. Thus the authors are considering a situation where, by construction, the valley currents do not give any meaning. This does not remove the reason for doing the calculation, but it removes the foundations of all critical remarks made on using the valley currents as a vehicle of calculation. It also removes the hopes of resolving of some of the difficulties related to the interpretation/observation of valley currents. In my view the authors are presenting a calculation (maybe a model calculation) in a situation where valley currents cannot be defined - and this should be clear in the submitted manuscript.

Requested changes

1- remove "cristal" - it is crystal in English 2-Where does Eq.(11) come from - it is crucial for many subsequent developments. Either give a derivation, or a complete sequence of references. 3) I seem to recognize the expressions for the Berry curvature (e.g., Eq.(20)). Is the expression new, or a reincarnation of well-known results?

Recommendation

Ask for minor revision

  • validity: good
  • significance: good
  • originality: good
  • clarity: high
  • formatting: excellent
  • grammar: -

Author:  Tatiana Rappoport  on 2024-06-05  [id 4665]

(in reply to Report 1 on 2024-05-13)

We thank the referee for reviewing our manuscript, and for recommending it for publication with minor revisions. The pertinent comments and suggestions have contributed to improve the quality of our paper. In what follows we address the points raised and list the changes made in the revised version of our article, which we believe is now ready for publication.

Attachment:

scipost_202406_00012v1_resubmission_letter.pdf

---

## Editorial Decision

resubmitted